# Comparison of Prognostic Value of 10 Biochemical Indices at Admission for Prediction Postoperative Myocardial Injury and Hospital Mortality in Patients with Osteoporotic Hip Fracture

**DOI:** 10.3390/jcm11226784

**Published:** 2022-11-16

**Authors:** Alexander Fisher, Wichat Srikusalanukul, Leon Fisher, Paul N. Smith

**Affiliations:** 1Departments of Geriatric Medicine, The Canberra Hospital, ACT Health, Canberra 2605, Australia; 2Departments of Orthopaedic Surgery, The Canberra Hospital, ACT Health, Canberra 2605, Australia; 3Medical School, Australian National University, Canberra 2605, Australia; 4Department of Gastroenterology, Frankston Hospital, Peninsula Health, Melbourne 3199, Australia

**Keywords:** hip fracture, mortality, myocardial injury, biochemical markers, predictors

## Abstract

Aim: To evaluate the prognostic impact at admission of 10 biochemical indices for prediction postoperative myocardial injury (PMI) and/or hospital death in hip fracture (HF) patients. Methods: In 1273 consecutive patients with HF (mean age 82.9 ± 8.7 years, 73.5% women), clinical and laboratory parameters were collected prospectively, and outcomes were recorded. Multiple logistic regression and receiver-operating characteristic analyses (the area under the curve, AUC) were preformed, the number needed to predict (NNP) outcome was calculated. Results: Age ≥ 80 years and IHD were the most prominent clinical factors associated with both PMI (with cardiac troponin I rise) and in-hospital death. PMI occurred in 555 (43.6%) patients and contributed to 80.3% (49/61) of all deaths (mortality rate 8.8% vs. 1.9% in non-PMI patients). The most accurate biochemical predictive markers were parathyroid hormone > 6.8 pmol/L, urea > 7.5 mmol/L, 25(OH)vitamin D < 25 nmol/L, albumin < 33 g/L, and ratios gamma-glutamyl transferase (GGT) to alanine aminotransferase > 2.5, urea/albumin ≥ 2.0 and GGT/albumin ≥ 7.0; the AUC for developing PMI ranged between 0.782 and 0.742 (NNP: 1.84–2.13), the AUC for fatal outcome ranged from 0.803 to 0.722, (NNP: 3.77–9.52). Conclusions: In HF patients, easily accessible biochemical indices at admission substantially improve prediction of hospital outcomes, especially in the aged >80 years with IHD.

## 1. Introduction

Osteoporotic hip fracture (HF), one of the leading health problems in the geriatric population all over the world, is associated with serious postoperative complications and increased mortality (between 5.1% and 16.3% in the hospital, up to 36% within 1 year) [1,2,3,4,5,6]. Over the last four decades the mortality rates after HF surgery remain unchanged [7]. Cardiovascular diseases (CVDs), especially the ischaemic heart disease (IHD), are recognised as major contributors of poor outcome in patients undergoing major noncardiac surgery [8,9,10,11,12,13,14,15,16], including HF repair [17,18,19,20,21,22,23,24,25]. Perioperative cardiac complications substantially prolong hospitalisation and account between one third and one-half of all deaths [9,15,26,27,28]. Given the importance of perioperative evaluation, risk stratification and individualised management of patients undergoing non-cardiac surgery a set of clinical practice guidelines were developed by the Joint Task Force of the European Society of Cardiology (ESC) and European Society of Anaesthesiology (ESA) [8] and the American College of Cardiology (ACC) and the American Heart Association (AHH) [29].

Many prognostic models to estimate preoperatively the probability of developing in-hospital major adverse cardiac events (MACE) in patients undergoing noncardiac surgery have been proposed and the widely acknowledged: The American Society of Anaesthesiologists Physical Status [ASA-PS], The Revised Cardiac Risk Index [RCRI)], The Universal American College of Surgeons NSQIP surgical risk calculator [ACS NSQIP], Preoperative Score to Predict Postoperative Mortality (POSPOM), Clinical Frailty Scale [CFS]. However, some of these models rely on subjective clinician judgment (ASA score), other did not adjust for possible confounders, most of the scores require extensive information regarding comorbidities and laboratory data (e.g., 21 patient-specific variables for ACS NSQIP [30] and summarising the data on individual patients may be difficult (need use of calculators [31,32]). Most importantly, these prognostic models (based in total on 26 scores [33]), although effective on the population level [34], when applied to an individual patient quite often result in inconclusive predictions [35]. The RCRI score, for example, underestimated 50% of adverse events [36,37]. In non-cardiac surgery patients, the POSPOM showed good discriminatory performance, but poor calibration with an overestimation of in-hospital mortality as well as poor performance for prediction of postoperative complications [38,39].

The predictive value of several clinical scoring systems proposed for assessment outcome in HF patients is currently debated [5,40,41]; none of the existing models yielded excellent discrimination while some models had a lack of fit [42]. Despite extensive studies have been carried out the problem of risk stratification and predicting HF outcomes at admission is far from being solved. The vast majority of models predicting HF outcomes are based on demographics, comorbidities and rarely include few blood marker, the proposed preoperative patient risk scores remain imperfect [41] even when clinical, demographic and inflammatory indices are combined [43].

In recent years, a number of different biochemical dysregulations, come to light not only as conditions associated old age, systemic chronic diseases, including IHD, osteosarcopenia, falls and fractures, but as main factors contributing to postoperative complications and all-cause mortality. Converging evidence increasingly implicates that several serum biomarkers predict poor outcomes in different settings (such as surgery, trauma, sepsis, various cancers, etc.). However, research regarding biochemical predictors in HF is limited, only few studies have concentrated on 1–2 biomarkers of a single outcome and, not surprisingly, biochemical indices still are rarely used.

In this study we attempted to assess and compare the usefulness (incremental value) of measurement of routine biochemical parameters before HF surgery and identify which indices are best to predict postoperative myocardial injury (PMI) and/or in-hospital death. Among most often reported in the literature biochemical predictive indices we have chosen for evaluation 10 biomarkers focusing on easily obtained parameters known to be associated with chronic systemic diseases, in particular CVDs/IHD and osteoporosis, and being factors allowing modification. The battery of these serum biomarkers included: parathyroid hormone (PTH) [44,45,46,47,48,49,50,51,52,53,54,55], vitamin D [51,56,57,58,59,60,61,62,63], albumin, alanine aminotransferase (ALT), gamma-glutamyl transferase (GGT) [64,65,66,67,68,69,70,71,72,73,74,75,76,77,78], Urea [79,80,81,82,83,84,85,86,87,88,89,90], GGT/ALT ratio [51,91,92,93,94,95], GGT/Albumin ratio [96,97], Urea/Albumin ratio [98,99,100,101,102] and Platelets/Albumin ratio [103,104].

These biomarkers have not been systematically investigated to predict in HF patients at admission the risk of development postoperative myocardial injury (PMI) or death. No studies have compared the prognostic value of different indices, the proposed cut-off levels of which differed significantly, and the interpretation of the results was controversial (for example, ALT and mortality [76,105,106,107,108]). Thus, it remains to be established which on-admission biochemical indicators may be useful prognostic biomarkers in HF patients.

## 2. Patients and Methods

### 2.1. Participants

In total 1273 consecutive patients (older than 60 years) admitted with a low-trauma non-pathological HF (cervical or trochanteric) to the Department of Orthopaedic Surgery of the Canberra Hospital (tertiary university centre) between 2010 and 2019 and had operative fracture treatment were included in the study. Inclusion criteria were as follows: (1) a definite diagnosis of hip fracture (intracapsular [cervical] or trochanteric) by imaging, (2) surgical HF repair, (3) age ≥ 60 years, (4) complete clinical and laboratory data. Exclusion criteria were: (1) subtrochanteric fracture, (2) medium- or high- energy trauma fracture (fall from height, car accident, etc.); (3) multiple fractures or polytrauma, (4) pathological fracture (malignant tumour)” The mean age of patients was 82.9 ± 8.7 [SD] years, 73.5% were women, and 50.5% had a cervical fracture. All patients followed a similar postoperative protocol with mobilisation out of bed on day one and urinary catheter out on day two.

The validation cohort (*n* = 582, mean age 81.9 ± 9.13 years, 71.0% women, 52.9% with cervical fracture) included patients admitted after those in the derivation group; they had a similar to the main cohort profile of chronic comorbid diseases, admission laboratory characteristics and outcomes.

### 2.2. Data Collection

In all patients the previous hospital and general practitioners’ medical case records were reviewed, data on socio-demographic (including pre-fracture residential status, use of walking aid), lifestyle factors (smoking status, alcohol use), clinical (12 chronic comorbidities, medications used) and laboratory parameters (21 variables) at admission and the hospital outcomes were prospectively recorded and analysed.

### 2.3. Laboratory Measurements

The routine laboratory tests included full blood count, serum electrolytes, creatinine, urea nitrogen, C-reactive protein (CRP), albumin and liver function tests, cardiac troponin I (cTnI), 25(OH) vitamin D (25(OH)D), intact PTH, thyroid stimulatory hormone (TSH), free thyroxine (T_4_), vitamin B_12_, folic acid, iron, ferritin, and transferrin. All these analyses were performed on the day of blood sampling (usually within 12–24 h after arrival at the Emergency Department) by standard automated laboratory methods. Serum calcium concentrations were corrected for serum albumin, glomerular filtration rate was estimated (eGFR). Serum cTnI and CRP levels were also assessed within 24 h post-operatively and then after if elevated and/or clinically indicated. All patients with elevated cTnI level of >20 ng/L or greater (“abnormal” laboratory threshold) were assessed for ischaemic features (ischaemic symptoms and 12-lead electrocardiogram).

### 2.4. Definitions

Chronic kidney disease (CKD) was defined as eGFR < 60 mL/min/1.73 m^2^, which was calculated by the Modification of Diet in Renal Disease (MDRD) equation. Anaemia was defined as haemoglobin level < 130 g/L in men and <120 g/L in women. Vitamin D deficiency was defined as 25(OH) D < 25 nmol/L and hypovitaminosis D as < 50 nmol/L, hyperparathyroidism was defined as elevated serum PTH (>6.8 pmol/L, the upper limit of the laboratory reference range), hypoalbuminaemia—as <33 g/L (lower level of reference range). The cut-offs for other biochemical variables were as follows: for urea—7.5 mmol/L (upper limit of reference range), ALT ≥ 17 IU (median value), for GGT ≥ 26 IU (median value), for GGT/ALT > 2.5, for GGT/Albumin ≥ 7.0, for Urea/Albumin ≥ 2.0, for Platelets/Albumin ≥ 5.9. The robustness of the aforementioned cut-offs was validated in our prior studies [17,45,47,51,109,110].

### 2.5. Outcome Measures

These included: (1) postoperative myocardial injury (PMI) defined by cTnI rise (if at least one of postoperative cTnI measurement values was >20 ng/L on days 1–5 post surgery with or without associated ischemic symptoms); (2) high inflammatory response assessed by marked elevation of CRP (>100 mg/L after the 3rd postoperative day); (3) length of hospital stay (LOS); (4) all-cause in-hospital mortality.

Most HF patients are not candidates for and did not have a coronary angiogram and there are no general guidelines or thresholds for acceptable cTnI elevations after various non-cardiac surgical procedures [111]; therefore, in this study, postoperative new acute myocardial infarction (AMI) was defined by cTnI ≥ 500 ng/L (25 times above the upper limit of reference levels) accompanied by obvious ECG signs (Q-waves, ST-segment changes, T-wave inversion) indicative of myocardial infarct.

### 2.6. Ethical Approval

The study was performed in accordance with the Declaration of Helsinki (1964) and its later amendments (as revised in 2013). The study was approved by the Australian Capital Territory Research Ethics Committee (ETHLR.18.085; REGIS Reference 2020/ETH02069).

### 2.7. Statistical Analyses

Data analyses were carried out using Stata software version 16 (StataCorp, College Station, TX, USA). Continuous variables were reported as means ± SD and categorical variables as percentages. Comparisons between groups were performed using analysis of variance and Student’s *t*-test for continuous variables and χ^2^ test (Yates corrected) for categorical variables. Univariate and multivariate (both linear and logistic) regression analyses were used to determine the odds ratio (OR) and 95% confidence intervals (CI) for associations between an outcome (dependent variable) and different clinical and laboratory variables; all potential confounding variables with statistical significance ≤ 0.15 on univariate analyses were included in the final multivariate analyses. Receiver operating characteristic (ROC) curve analysis (the area under the ROC curve, AUC) was used to investigate the discriminatory power of preoperative indices to predict postoperative events. Sensitivity, specificity, accuracy, positive predictive value (PPV), negative predictive value (NPV), positive likelihood ratio (LP+), negative likelihood ratio (LP−) and number of patients needed to be examined for correct prediction (NNP [112,113]) were calculated to assess the discriminatory performance of the tests. The predictive performance of the models was further assessed using goodness-of-fit statistics for calibration by Hosmer-Lemeshow test. All tests were two-tailed; *p*-values < 0.05 were considered statistically significant.

## 3. Results

### 3.1. Baseline Characteristics and Outcomes

Of 1273 patients who underwent HF surgery during the study period, 361 (28.4%) have been previously diagnosed with IHD and 99 subjects (7.8% of the total cohort, 27.4% among IHD patients) had a history of AMI. Sociodemographic data, comorbidities, and outcomes in patients with and without IHD are presented in Table 1. The cohort was almost equally split with cervical (50.7%) and trochanteric (49.3%) HFs, and there was no significant difference between groups concerning surgery and anaesthesia. Patients with IHD were significantly older (+2.7 years on average), were less likely to be female and alcohol over-users, had a higher prevalence of hypertension, CKD, chronic obstructive pulmonary disease (COPD), cerebrovascular accident (CVA), type2 diabetes mellitus (T2DM) and Parkinson’s disease (for two last diseases *p* = 0.058), and more often used walking aids. The percentage of ex-smokers, permanent residential care facilities (PRCF) residents and patients with dementia, anaemia and TIA did not differ in these two groups. The total all-cause in-hospital mortality was 4.8%, in patients without IHD—3.7%, with IHD—7.5% (contrast +3.8%, *p* = 0.005), and among those with previous AMI—11.8% (contrast +8.1%, *p* < 0.001). Patients with IHD, as would be expected, more often developed PMI (58.6% vs. 37.7%, *p* < 0.001), AMI (11.7% vs. 4.8%, *p* < 0.001), a high inflammatory response (CRP > 100 mg/L in 84.2% vs. 79.7%, *p* = 0.037) and had a prolonged hospital stay (LOS > 20 days in 25.8% vs. 20.4%, *p* = 0.024) compared with the non-IHD persons. IHD patients with a fatal outcome compared with survivors were older (+4.0 years: 88.6 ± 5.34 vs. 84.6 ± 7.23, *p* = 0.006), all but one > 80 years of age (96.3% vs. 75.1%, *p* = 0.006) and more often had CKD (70.4% vs. 44.0%, *p* = 0.007), while all other examined sociodemographic (including male sex prevalence: 9.1% vs. 6.8%, *p* = 0.285) and comorbid characteristics did not show statistical difference among the groups. In HF patients, presence of IHD increases the risk of a fatal outcome (after controlling for age, gender, HF type, preoperative residence, mobility status, comorbidities) by twofold (OR 2.1, 95% CI 1.24–3.51, *p* = 0.005), of developing PMI by 2.3-fold (OR 2.3, 95% CI 1.81–3.01, *p* < 0.001) and a postoperative AMI by 2.4-fold (OR 2.4, 95% CI 1.98–4.02, *p* < 0.001).

PMI occurred in 555(43.6%) patients (Table 2). Compared to the rest of the cohort, patients with PMI, not surprisingly, were older (+5.3 years), more often >80 years of age (85.2% vs. 60.4%, *p* < 0.001), males (28.9% vs. 24.6%, *p* = 0.054), PRCF residents (38.7% vs. 28.4%, *p* < 001), more frequently had a history of IHD (37.7% vs. 20.6%, *p* < 0.001), AMI (11.1% vs. 5.2%, *p* = 001), hypertension (60.2% vs. 51.4%, *p* = 0.001), TIA (12.6% vs. 8.3%, *p* = 0.009), anaemia (46.0% vs. 38.4%, *p* = 0.005) and dementia (38.5% vs. 26.2%, *p* < 0.001), but less likely were alcohol over-users (1.9% vs. 5.4%, *p* = 0.025), current smokers (4.1% vs. 6.5%, *p* = 0.068) or suffered from Parkinson’s disease (3.8% vs. 5.9%, *p* = 0.052); history of stroke, COPD, T2DM, ex-smoking and use of walking aids were not associated with PMI. Patients who developed PMI (irrespective of IHD history) had a significantly higher mortality rate (8.8% vs. 1.9%, contrast +6.9%, *p* < 0.001); in the PMI group the proportion of individuals with high inflammatory responses (CRP > 150 mg/L in 69.2% vs. 55.1%, *p* < 0.001) and LOS > 10 days (61.4% vs. 54.9%, *p* = 0.013) was also markedly higher. Postoperative AMI experienced in total 6.7% of patients, including 11.7% with previously known IHD and 4.8% without IHD. Notable, PMI was observed most often in the first 1–3 days after surgery (when patients were receiving analgesic medications that can mask ischaemic symptoms) and was asymptomatic in 97.8% of these patients, symptomatic only in 15 individuals, including 9 with postoperative AMI; the injury probably would have gone undetected without routine cTnI measurements. Rise of cTnI occurred in 58.6% of patients with IHD (hospital mortality 12.9%), and in 62.1% of patients with a history of AMI (mortality 15.8%). Both IHD and PMI were also associated with a high inflammatory response and prolonged hospital stay (Table 1 and Table 2).

Clearly, in HF patients, both advanced age (>80 years) and presence of IHD are strong indicators of worse in-hospital outcomes, in particular, PMI and mortality which are interconnected. Among patients who died 56 (91.8%) were aged >80 years, 27(42.3%) had a history of IHD (including previous AMI in 9 [33.3%]), and 48 (78.7%) experienced PMI. In the group who developed PMI, 37.7% of patients were previously diagnosed with IHD and most were aged >80 years (85.2%). As shown in Table 3, in HF patients, the risk of a fatal outcome is about 5-fold higher among the aged >80 years (OR 4.9 (1.95–12.33), twice as great in subjects with a history of IHD compared with those without IHD (OR 2.1, 95% CI 1.24–3.51, *p* = 0.005), and 5-fold higher among patients who developed PMI (OR 5.0, 95% CI 2.70–9.41, *p* < 0.001). Furthermore, in subjects aged >80 years with a history of IHD the risk of PMI is 8.3 times higher (OR 8.3, 95% CI 5.58–12.36, *p* < 0.001) and risk of a lethal outcome is 7.4 time higher (OR 7.4, 95% CI 2.55–21.51, *p* < 0.001) compared with HF patients without such characteristics.

These findings, which are in line with intuitive expectations and in accordance with results of previous studies, once again underscore the importance and utility of advanced age, history of IHD and developing PMI for elucidating the prognosis and identifying the highest risk groups. On the other hand, it should be emphasised that most of aged HF patients even with known IHD and/or PMI (including new AMI) survive suggesting the need of better more precise prediction tools, especially for individuals with abovementioned characteristics.

### 3.2. Prognostic Role of Biochemical Indices at Admission

As the odds for a fatal in-hospital outcome in patients with IHD is twice as great compared with those without IHD and near half (43.6%) of all HF patients postoperatively demonstrate some degree of myocardial injury (a complication which, in turn, increases the risk of in-hospital death by 5-fold), a crucial question should be introduced: how to determine at time of admission which patients (especially among those with a history of IHD), are at the highest risk of PMI and hospital death?

In this study, based on our previous data (including multivariate logistic regression) and literature reports on the predictive capability of different biochemical parameters we evaluated and compared the potential of 10 routine preoperative metabolic indices for predicting adverse events in HF. We analysed the prognostic value and prediction performance each of the 10 metabolic indicators separately and in combination with history of IHD or/and advanced age. The obtained estimates for the total cohort, for patients with pre-HF known IHD, and for IHD patients aged >80 years are shown in Table 3 and Table 4; the data are presented in rank order from highest to lowest ORs and AUCs.

The impact of different preoperative metabolic characteristics as prognosticators for fatal hospital outcome was as follows (Table 3). Whereas history of IHD doubled the risk of a fatal hospital outcome, when one of the studied biomarkers (except GGT > 26 IU and ALT > 17 IU) was added to the model for in-hospital death the ORs ranged between 2.6 and 8.0 (being 3.3 and above for 6 biomarkers). IHD in patients aged > 80 years was associated with a 7.4-times higher risk of postoperative death, while adding any of the biochemical indicators, except ALT > 17 IU, showed in this group ORs between 7.7 and 23.6 (9.2 and above for 7 variables).

Testing the biochemical status at admission improved also significantly the prognostication of PMI in IHD patients, particularly in the aged > 80 years (Table 3). The risk of PMI in patients with IHD was 2.3 times higher than in non- IHD and increased to 3.1 in IHD patients with elevated PTH, to 3.9 in cases with vitamin D deficiency, to 4.4 if urea > 7.5 mmol/L, and to 4.5 if Urea/Albumin ratio ≥ 2.0. Age above 80 years was associated with a 3.8-fold increased risk for developing PMI; in the aged IHD patients the OR for PMI was 8.3 and raised to 9.1(in subjects with admission GGT ≥ 26 IU), to 9.9 (if GGT/Albumin ratio ≥ 7.0), to 10.4 (if vitamin insufficiency present), to 11.4 (if GGT/ALT > 2.5), to 12.6 (if Urea/Albumin ≥ 2.0), to 12.8 (if urea > 7.5 mmol/L), and to 18.0 (in cases with vitamin D deficiency).

Whereas IHD and advanced age alone demonstrated a fair prognostic ability for mortality and PMI, the on-admission biochemical parameters add net benefit showing clinical usefulness, especially when patients were stratified into groups representing IHD and aged > 80 years. For example, PTH > 6.8 pmol/L at admission increased the risk of a fatal outcome in the total HF cohort by 1.9-fold, the risk of developing PMI by1.3-fold, in patients with IHD by 3.7- and 3.1-fold, respectively, and among individuals with IHD aged > 80 years by 11.8- and 8.3-fold, respectively. Similarly, admission urea > 7.5 mmol/L indicated a 2.2-fold higher risk of lethal outcome in the total HF cohort, 5.1-fold higher risk among IHD patients and 9.5-fold higher risk in the aged IHD group (Table 3). Some biomarkers (25(OH)D < 50 mmol/L, Albumin < 33 g/L, and GGT/ALT > 2.5) did not show prognostic value when analysed in the total HF cohort but demonstrated a significant prognostic effect in IHD patients (ORs of 2.7, 3.3 and 2.6, respectively), especially in the aged with IHD (OR of 8.4, 15.2 and 8.5, respectively). These observations indicate that metabolic parameters even prognostically not significant for the total HF cohort are useful when clinical characteristics are taken into account. It is worth to note that in our study most preoperative biochemical variables and their ratios were in the “normal range” or only mildly elevated and commonly have been considered as non-diagnostic and non-prognostic. It appears that in aged HF patients with IHD presence on arrival at least one of abovementioned biomarkers increases the risk of a fatal outcome or PMI by 3–4 times.

### 3.3. Predicting Performance of Biochemical Indices at Admission

In the total HF population, multivariate logistic regression model (included all variables associated with in-hospital death on univariate analysis) revealed as significant independent predictors of mortality at admission the following: age > 80 years, PTH > 6.8 pmol/L, urea > 7.5 mmol/L and GGT/Albumin ≥ 7.0; this model yielded AUC of 0.725 (95% CI 0.663–0.788). The multivariate logistic regression for hospital death based on the same approach and development of PMI (postoperative cTnI rise) demonstrated in addition to abovementioned characteristics as independent predictors also vitamin D < 25 mmol/L and PMI; this model improved the prediction a fatal outcome and yielded AUC of 0.767 (95% CI 0.710–0.824). The independent preoperative predictors of PMI (in the total HF cohort) were history of IHD, age > 80 years, PTH > 6.8 pmol/L, urea > 7.5 mmol/L and male sex, AUC 0.700 (95% CI 0.671–0.730).

Next, we evaluated the predictive performance of the biochemical parameters at admission for in-hospital mortality or/and developing PMI. Table 4 provides the summary of discrimination ability (AUC and related characteristics) and calibration of the tests for IHD patients aged >80 years, the group with the highest risk of poor outcomes. Although in this group all evaluated biomarkers demonstrated an increased OR for mortality or/and PMI, not all of them had a reasonable predictive performance.

For predicting mortality, models based on the clinical characteristics (age > 80 years, presence of IHD), or GGT ≥ 26 IU, or ALT ≥ 17 IU, or combination of these variables demonstrated low-modest values for AUC (0.591–0.700), while in other tests/models AUC ranged from 0.803 to 0.722. Metabolic indicators when added to two clinical characteristics (IHD, age > 80 years) significantly improved the prediction of postoperative mortality: AUC increased from 0.591 (for IHD) or 0.637 (for age > 80 years) to 0.803 (if 25(OH)D < 25 mmol/L), to 0.789 (if Albumin < 33 g/L), to 0.742 (if GGT/ALT > 2.5), to 0.729 (if Urea > 7.5 mmol/L) and to 0.725 (if PTH > 6.8 pmol/L). The sensitivity and specificity as well as PPV, NPV and other performance parameters of different biomarkers varied broadly (as expected, less-sensitive variables being more specific). Four tests have a very good sensitivity above 90% (90.9% to 94.1%), and 5 other tests have sensitivity of 80% and above (80.0–88.2%). On the contrary, the majority of tested models (7) have a specificity slightly above 50% (50.6–58.7%), in 3 tests it ranged from 68.6% to 72.4%, one variable (age > 80 years) was not specific (30%), and only two indices—albumin < 33 g/L and 25(OH)D < 25 mmol/L- have a specificity of 84.6% and 90.3%, respectively. Accordingly, the positive predictive values (PPV) of the tests were quite low (ranging from 6.2% to 28.1%) but the negative predictive values (NPV) were very good (96.8%-99.3%). All models, except one (with 25(OH)D < 50 mmol/L), showed appropriate calibration: Hosmer-Lemeshow goodness-of-fit test consisted of 0.17–14.82 (*p* value ranged between 0.1908 and 0.9941), in the poorly calibrated model it was 20.39 (*p* = 0.0401). In our cohort, the predictive performance of three ratios—GGT/Albumin, Urea/Albumin and Platelet/Albumin ratio was not superior (or even slightly lower) to that of serum albumin (<33 mg/L) and Urea (>7.5 mmol/L) alone.

A helpful and practical criterion to compare the prediction validity (robustness) of the studied models is the number of patients with a given condition (s) who need to be examined in order to detect/correctly predict one person with a certain outcome (number needed to predict, NNP). The NNP a fatal outcome in HF patients based only on the presence of IHD was 26.3, based only on age > 80 years was 20.4, and on combination of both characteristics was 12.5. The NNP decreased dramatically when biochemical parameters at admission were taken into account: 25(OH)D < 25 mmol/L (NNP = 3.8), GGT/ALT > 2.5 (NNP = 4.8), Platelets/Albumin ≥ 5.9 (NNP = 5.2), albumin < 33 g/L(NNP = 6.8), and 25(OH)D < 50 mmol/L (NNP = 7.1); use of other biochemical markers showed NNP between 9.2 and 10.6. These data further confirm the prognostic usefulness of biochemical characteristics at admission.

To sum, for predicting at admission in-hospital mortality in HF patients with IHD aged >80 years the best values showed five models (based on vitamin D deficiency/insufficiency or Albumin < 33 g/L or GGT/ALT > 2.5 or Urea > 7.5 mmol/L or PTH > 6.8 pmol/L). Presence of vitamin D deficiency (25(OH)D < 25 mmol/L) had the largest AUC (0.803), the highest predictive accuracy (90.3%), NPV (98.4%), LR+ (7.946), LR– (0.337) and the lowest NNP (3.8). A high discrimination performance demonstrated also four other models comprised of Albumin < 33 g/L (AUC 0.789, accuracy 84.6%, NPV 98.8%, NNP 6.8), or GGT/ALT > 2.5 (AUC 0.742, accuracy 70.2%, NPV 98.4%, NNP 4.8), or urea > 7.5 mmol/L (AUC 0.729, accuracy 60.4%, NPV 98.5%, NNP 9.5), or PTH > 6.8 pmol/L (AUC 0.725, accuracy 56.4%, NPV 98.9%, NNP 9.2). In terms of sensitivity three biomarkers (25(OH)D < 25 mmol/L [69.2%], Albumin < 33 g/L [72.7%], GGT/ALT > 2.5 [78.6%]) are inferior to PTH > 6.8 pmol/L [90.9%] but have higher specificity (91.3%, 85.1%, 69.8%, vs. 54.1%, respectively). These indicate that in the heterogeneous HF cohort different biomarkers possible identify subgroups of patients with specific or more pronounced metabolic changes; therefore, using simultaneously different indices would further benefit the prediction decision.

Of 61 patents who died in the hospital, 56 (91.8%) subjects at admission had at least one of these five indicators, 48 (78.7%) patients (96.3% among IHD patients) had two or more and 26 (42.6%) had three or more biomarkers, while only in 5 (8.2%) patients, including one (3.7%) with IHD, none of these biomarkers was found. In other words, the on-admission prediction by biochemical characteristics for a fatal outcome in the total cohort of HF patients was consistent with the actual observation.

Regarding the preoperative prediction of PMI, 5 tests at admission (urea > 7.5 mmol/L; Urea/Albumin ≥ 2.0, GGT/ALT > 2.5, 25(OH)D < 50 mmol/L and GGT/Albumin ≥ 7.0) showed a good discriminative performance with values for AUC ranging between 0.782 and 0.755, and 5 more tests have modest AUC values (between 0.742 and 0.711). Sensitivity above 80% (80.7–87.7%) demonstrated 4 tests, between 70% and 80% (71.0–79.2%) other 5 tests, whereas 2 tests (25(OH)D < 25 mmol/L and IHD) showed low sensitivity (36.1% and 37.7%, respectively). Specificity above 90% exhibited 2 tests (25(OH)D < 25 mmol/L [97.0%] and albumin < 33 g/L [90.7%]), between 80% and 90%—2 tests, and between 70% and 80%—9 tests, while age > 80 years was not specific (39.6%). A PPV above 70% have only 2 tests, and the NPV was above 80% in 11 tests. Values for the likelihood (LR+) of PMI to be predicted by these biomarkers were high (range 11.9–2.5) suggesting balance in favour of wright conclusion over misdiagnosis. All models were well calibrated. Of note, urea alone had a slighter larger AUC compared with the Urea/Albumin ratio (0.782 vs. 0.780), the AUC for GGT/ALT ratio (0.760) was larger than that for GGT/Albumin ratio (0.755) but the Platelet/Albumin ratio was inferior to that of IHD in aged >80 years (0.711 vs. 0.741).

As shown in Table 4, the NNP the development of PMI based only on IHD history was 4.8, based on advanced age was 3.4, on both characteristics was 2.2. The NNP decreased below 2.0 by adding one of the following biochemical parameters: urea > 7.5 mmol/L (NNP = 1.6), 25(OH)D < 25 mmol/L (NNP = 1.6), GGT ≥ 26 IU (NNP = 1.8), Urea/Albumin ≥ 2.0 (NNP = 1.8), GGT/ALT > 2.5 (NNP = 1.9), 25(OH)D < 50 mmol/L (NNP = 1.96).

The first 5 places in terms of the weight of the pre-operative prediction PMI in the aged IHD patients (the greatest AUCs) occupied the models with the following indices: Urea > 7.5 mmol/L (AUC 0.782, accuracy 82.2%, NPV 91.3%, NNP 1.6), Urea/Albumin ≥ 2.0 (AUC 0.780, accuracy 77.8%, NPV 84.5%, NNP 1.8), GGT/ALT > 2.5 (AUC 0.760, accuracy 79.0%, NPV 84.2%, NNP 1.9), 25(OH)D < 50 mmol/L (AUC 0.757, accuracy 77.7%, NPV 83.0%, NNP 1.96), and PTH > 6.8 pmol/L (AUC 0.742, accuracy 73.7%, NPV 81.3%, NNP 2.1).

These biomarkers demonstrated also high predictive performance for PMI in the total HF cohort. Among 555 patients who developed PMI, 500 (90.1%) had at least one of the abovementioned five biomarkers, 396 (71.4%) patients had two or more such characteristics including 235 (42.3%) subjects with ≥3 biomarkers; only in 55 (9.9%) of patients with PMI none of the five biomarkers presented at admission; these 5 biomarkers were able to predict PMI in most cases (in 500 among 555 actually observed).

Notable, four on-admission biomarkers—GGT/ALT > 2.5, PTH > 6.8 pmol/L, vitamin D deficiency/insufficiency, and Urea > 7.5 mmol/L—were the best at admission predictors for both PMI and hospital death, confirming, as would be expected, commonality of risk factors and underlying pathophysiological mechanisms.

Next it was important to determine which (if any) preoperative factors increase the risk of death among individuals who developed PMI (regardless history of IHD). Our analysis revealed that in the group with PMI, most likely to progress to a lethal outcome were patients with the following on-admission characteristics: urea > 7.5 mmol/L (74.5% vs. 53.6%, *p* = 0.006), PTH > 6.8 pmol/L (71.7% vs. 52.6%, *p* = 0.013), urea/albumin ≥ 2.0 (78.7% vs. 56.8%, *p* = 0.004), Platelets/albumin ≥ 5.9 (67.4% vs. 45.3%, *p* = 0.004), GGT/Albumin ≥ 7.0 (63.8% vs. 48.4%, *p* = 0.044), 25(OH)D < 25 mmol/L (15.8% vs. 10.0%, *p* = 0.039) and CKD (59.6% vs. 42.8%, *p* = 0.027). These observations emphasise the role of shared physiological pathways and biochemical dysregulations as factors underlying adverse postoperative outcomes in HF patients.

Taken together, our results demonstrated that preoperative biochemical indices are helpful and can significantly improve the discrimination capability of the clinical factors (e.g., advanced age, IHD, CKD) for predicting worse hospital outcomes. For example, if a HF patient with IHD had at admission vitamin D deficiency the risk for developing PMI or hospital death was 3.9 and 8.0 times higher, respectively, than in those without such characteristics, and if this subject was >80 years old the corresponding figures were 18.0 and 23.6. Similarly, the risks of PMI or/and death in HF patients with IHD and Urea > 7.5 mmol/L at admission were 4.4- and 5.1-fold higher, respectively, compared with individuals without these features, whereas in the IHD patients ≥ 80 year these risks were 12.8 and 9.5 times higher, respectively. Understandable, indices with high sensitivity and low specificity should be used for identifying patients with a high probability of poor outcomes, while tests with low sensitivity and high specificity may be considered as indicators for excluding poor outcomes (a poor outcome is unlikely).

In the total HF cohort, at arrival at least one of the 5 strongest predictive biochemical indices was observed in 91.8% of all subjects with a fatal outcome and in 90.1% of all patients who developed PMI. In patients with IHD, presence of these biochemical parameters at admission increased the risk of PMI and/or death by 2–3.5–fold. Neither presence of IHD, no age > 80 years per se are predictive of PMI and/or a fatal outcome if these clinical characteristics are not combined with at least one of the following metabolic abnormalities: vitamin D insufficiency/deficiency, hypoalbuminaemia, hyperparathyroidism, GGT/ALT > 2.5, Urea > 7.5 mmol/l, or Urea/Albumin ≥ 2.0. In other words, these factors associated with coexisting chronic diseases reflect significant pathophysiological conditions contributing to adverse outcomes and, therefore, may be complementary for prediction decisions in HF.

Importantly, the abovementioned metabolic dysregulations, are not only strongly indicative of poor outcomes in HF patients but are potentially modifiable and, therefore, should be diagnosed and addressed before a fracture occur. Can perioperative interventions aimed to normalise these metabolic disturbances improve HF outcomes need to be investigated.

### 3.4. Internal Validation

In the validation cohort, sociodemographic characteristics and the comorbidity profile, including the proportion of patients with IHD (28.3%), were not significantly different from that in the derivation group, PMI was observed in 44.1%, and the all-cause mortality rate was 4.1%. The on-admission biochemical parameters in both cohorts produced, in general, similar prognostic and predictive values. For example, risk of hospital death in IHD patients aged >80 years with Urea > 7.5 mmol/L was 9.2 (95% CI 1.89–14.82) times higher than in patients without such signs, AUC = 0.749 (95% CI 0.614–0.884), NNP 8.7; in subjects with GGT/ALT > 2.5 the corresponding figures were: OR 10.2 (95% CI 1.03–17.47), AUC 0.761 (95% CI 0.514–1.000), NNP 11.5; in case of Albumin < 33 g/L the corresponding figures were: OR 16.3 (95% CI 1.62–16.67), AUC 0.797 (95% CI 0.551–1.000), NNP 7.0; and if 25(OH)D was <50 mmol/L the corresponding figures were: OR 27.4(3.22–33.65), AUC 0.836 (95% CI 0.708–0.964), NNP 4.7. The AUC for prediction a lethal outcome by the 5 strongest biochemical indices ranged between 0.803 and 0.725 in the derivation, and between 0.836 and 0.749 in the validation cohort. All patents who died in the hospital had at admission at least one of the five indicators mentioned in derivation cohort, 82.6% subjects had two or more, and 47.8% had three or more such biomarkers. The Hosmer-Lemeshow statistic confirmed a good calibration of the models. In the validation cohort there were no lethal cases among the non-IHD group younger than 80 years with a normal PTH level, and there were no cases of PMI among individuals with GGT/Albumin ratio < 7 at admission.

The tested biochemical indices showed also a reasonable/good predictive performance for developing PMI, and the calibration was acceptable. As an example, in patients aged > 80 years and elevated PTH compared with subjects without these characteristics, the risk of PMI was 8.9-fold higher (95% CI 3.92–20.50), AUC 0.749 (95% CI 0.672–0.826), NNP 2.17; for Urea > 7.5 mmol/L the corresponding figures were: OR 10.2 (95% CI 4.33–24.14), AUC 0.756 (95% CI 0.672–0.841), NNP 2.24; for GGT/ALT > 2.5 the corresponding figures were: OR 9.0 (95% CI3.52–23.12), AUC 0.689 (95% CI 0.608–0.770), NNP 2.01. For the prediction of PMI, the AUC of the 5 best indicators ranged between 0.782 and 0.742 in the main cohort, and between 0.756 and 0.689 in validation cohort. In the group of patients who developed PMI, 88.6% had at least one of the five strongest biomarkers described in the derivation cohort, 58.2% patients had two or more such characteristics and in 34.7% subjects ≥ 3 biomarkers were found; however, in 11.4% of patients who experienced PMI none of these five biomarkers have been presented at admission suggesting an important role of some other factors responsible for this complication.

Taken together, internal validation showed satisfactorily calibrated prediction models, capable of predicting PMI and/or hospital death; these data confirm the prognostic usefulness (discrimination accuracy) and certainty of evidence based on consideration of abovementioned biochemical characteristics at admission.

## 4. Discussion

### 4.1. Main Findings

In HF patients, measurement of routine biochemical parameters at admission can significantly improve prognostication, risk stratification and individualised care, especially in persons of advanced age (>80 years) and/or with IHD, groups who are at the highest risk of poor outcomes (Figure 1). Among 10 biochemical indices evaluated and internally validated the best performance for predicting PMI and/or hospital death showed the following: GGT/ALT > 2.5, PTH > 6.8 pmol/L, Urea > 7.5 mmol/L, vitamin D deficiency (25(OH)D < 25 nmol/L) or insufficiency (25(OH)D < 50 nmol/L), Albumin < 33 g/L (for a fatal outcome) and Urea/Albumin ≥ 2.0 (for PMI). To the best of our knowledge, to date, no comprehensive comparative analysis of predictive values of biochemical indices at admission in HF patients has been performed, and none of the earlier studies have investigated any of these biomarkers to predict risk of development PMI.

This study suggested 4 important facts critical to osteoporotic HF management.

First, it confirmed that both age > 80 years and history of IHD are the strongest clinical factors indicative of worse short-term outcomes, namely, developing PMI and hospital death. Individuals aged >80 years accounted for 70.6% of the HF cohort, 76.7% among the IHD patients, 85.2% among subjects who developed PMI, and 91.8% in the fatal group. Patients with IHD accounted for 28.4% of the total HF cohort and 44.3% of all hospital deaths (mortality rate 7.5% vs. 3.7% among non- IHD subjects). PMI occurred (typically without chest pain or any ischemic symptoms) in 43.6% of HF patients (in 58.6% of individuals with IHD and in 37.7% without IHD). PMI was observed in 80.3% [49/61] of all deaths; the mortality rate among patients with PMI was approximately 4.6-fold higher than in patients without such complication (8.8% vs. 1.9%). PMI was also associated with high inflammatory responses and prolonged LOS. PMI (diagnosed by even minor troponin elevations), which is multifactorial (perioperative stress, blood loss, hypoxaemia, release of various inflammatory cytokines, arrhythmia, vascular endothelial cell injury, etc.) and often occurs in patients without previously known IHD (62.0% in our cohort), is currently recognised as a specific and sensitive indicator of perioperative complications and mortality in patients undergoing noncardiac surgery including orthopaedic surgery [7,13,35,114] and HF repair [7,24,114,115,116,117,118,119]. Therefore, it is important to have a sufficiently high index of suspicion for PMI, to consider its risk and timely predict in subjects with and without history of IHD.

Second. Although the prognostic role of advance age, presence of IHD and developing PMI is unquestionable, these factors, however, are not sufficient for individual predictions (i.e., identification of patients imminent for poor outcomes). In fact, only a small proportion of the “high-risk patients” demonstrated poor outcomes after HF: the majority of aged (>90%) or subjects with IHD (92.5%) or who developed PMI (91.2%) survived. Therefore, the crucial questions originated—how to determine at time of admission which aged HF patient, especially with IHD, is really at the highest risk of an adverse outcome (e.g., PMI and/or in-hospital death), in whom PMI (irrespective of presence underlying chronic diseases) should be expected and preventive measures applied. Obviously, the answers to these questions are enormously difficult because the pathogenesis of poor outcomes in HF is complex and multicausative. The outcomes are influenced by numerous interacting patient-related (age, gender, comorbidities), pre-, intra- (type of procedure, blood loss, hypotension, etc.) and postoperative (PMI, infection, delirium, acute kidney failure, etc.) factors—all of which may naturally increase the risks. In an attempt to come closer to a practical solution of the problem we evaluated 10 biochemical indices potential for predicting the occurrence of adverse postoperative events, focusing on factors allowing modification. We found that despite significant complexity and heterogeneity in clinical profile of HFs simple preoperative metabolic indices (even in supposedly “normal” range) provide significant prognostic information, substantially improving prediction of adverse outcomes, particularly among aged patients with IHD (the groups at the highest risks), and, possibly, may serve as important clues for treatment to reduce poor outcomes. Metabolic indicators integrate various genetic, lifestyle and environmental effects, reflect ongoing physiological processes in multiple organs, and, therefore, provide an insight into disease pathophysiology acting as powerful characteristics of an individual’s health status. Metabolic dysregulations are implicated in the pathogenesis and severity of numerous human pathologies, including osteoporosis, atherosclerosis, IHD (reflecting systemic connectivities), and, consequently, underly the adverse outcomes in HF patients. In HF patients, alterations in biochemical indices, IHD and advanced age interact to increase risks of developing PMI, new AMI, high inflammatory response, prolonged LOS and mortality above the effect of each individual condition, as demonstrated in this and our previous studies [17,45,51,110,120,121,122].

Our findings that on-arrival insufficient 25(OH)D levels and/or elevated PTH predict PMI and/or death are in line with long-standing evidence that the physiological effects of vitamin D and PTH extend beyond calcium homeostasis and bone mineralisation and consistent with many previous reports [17,123,124,125,126]. Hypovitaminosis D is also known to correlate with higher inflammatory response [124,127,128,129,130,131,132], which together with increased PTH levels (a factor associated with multiple postoperative complications in HF patients [45,133] and/or PMI significantly increases the risk of poor outcomes.

Third, comparison of predictive performance of 10 different biomarkers revealed 6 indices with the strongest discriminant ability. The most significant net benefit in predicting both developing PMI and/or in-hospital mortality demonstrated preoperative GGT/ALT ratio > 2.5, elevated urea (>7.5 mmol/L), hyperparathyroidism (PTH > 6.8 pmol/L) and abnormal vitamin D status (deficiency or insufficiency); preoperative Urea/Albumin ≥ 2.0 strongly indicated risk of PMI, and hypoalbuminaemia (<33 g/L)—a high risk of a lethal outcome.

The presented data clearly demonstrated that the long-held belief that IHD (even history of AMI) are significant risk factors for poor outcomes in HF patients is true predominantly when IHD is accompanied by metabolic disturbances such as elevated PTH, urea, GGT/ALT ratio, low albumin or 25(OH)D levels. Indeed, these biochemical characteristics were not presented only in 1 of 27 patients with IHD who died and in 50 (9.0%) of 555 patients who developed PMI. In other words, the prognostic performance of IHD depends on the underlying biochemical status. Certainly, risk stratification at admission, preventive and therapeutic interventions would be more precise if based not only on the history of IHD per se, but also considered the clinically silent metabolic profile, a key player in the development of comorbidities and worse outcomes. As mentioned above (Introduction) most of the routinely used in clinical practice approaches for predicting the risk of MACE in patients undergoing noncardiac surgery do not include biochemical parameters (except for serum creatinine) and are imperfect on individual level [36,37].

Reports on predictive capacity of several existing cardiac risk indices [134,135,136], RCRI scores in HF patients are scarce and often controversial [35,137]. Our data show that biochemical indices at admission may help to identify patients at high risk of cardiac events during and after surgery as well as to avoid unnecessary preoperative cardiac assessment. The concept that all IHD patents with HF have the same high risk of poor outcome appears no longer viable; predictions and therapies to be efficacious should address specific (including metabolic) risk factors of individual patients. The results presented here add new insights regarding the prognostic usefulness of preoperative biochemical indices and (if externally validated) may allow preoperative estimation of cardiac risk independent of direct risks caused by surgical and post-surgical complications. These appear as one step towards the patient-level precision to predict/prevent adverse outcomes early, an ideal goal which currently remains unmet.

Our simple models (based on three variables—IHD, age > 80 years and one biochemical index) for prediction a fatal outcome (AUCs: 0.803–0.722 in the main cohort and 0.836–0.749 in the validation cohort) or developing PMI (AUCs: 0.782–0.742 in the main cohort and 0.756–0.689 in the validation cohort) performed better than the model which included all independent variables in the total HF cohort (AUCs 0.725 and 0.700, respectively). Moreover, the simplified on-admission models for predicting hospital mortality after HF performed as well as or even better than most of earlier proposed models based on multiple variables (usually between 7 and 29) often including intra- and postoperative characteristics. To name for comparison a few with the highest discrimination ability: (1) 29 variables, AUC 0.91 [138]; (2) 29 variables, AUC 0.895 and 0.797 in training and testing datasets, respectively [139]; (3) >60 variables, AUC 0.83 (for 30 day mortality) and AUC 0.75 (for 1-year mortality) [140]; (4) 9 predictors, AUC 0.81 and 0.79 in the development and validation cohorts, respectively [141]; (5) 13 variables, AUC 0.82 [142]; (6) 7 variables, AUC 0.82 [143]; (7) 9 variables (Sernbo score), AUC 0.79 (for 1-year mortality) [144]; (8) >30 variables, AUC 0.76 [32]; (9) 29 variables, AUC 0.74 [145]; (10) 22 variables, AUC 0.71; (11) 7 variables AUC 0.71 (UK National Hip Fracture Database)-0.70 (Nottingham model) [146]; (12) 10 variables, AUC 0.702 [31]; (13) 9 variables (Sernbo score), AUC 0.79 (for 1-year mortality) [144]; (14) Charlson comorbidity index, AUC 0.682 (for in-hospital mortality [147], AUC 0.769 [148]–0.607 [149] (for 1-year mortality). A recent systematic review of prediction models for osteoporotic fractures (68 studies) found that in 69 of 70 models AUC ranged from 0.60 to 0.91 including only two models (3%) with AUC > 0.90 and 9 (13%) with AUC 0.8–0.9 [150].

Furthermore, in HF patients aged >80 years with a history of IHD, for correct prediction at admission of one PMI biochemical indices need to be assessed in 2 patients, and for correct prediction of one fatal outcome 4–7 subjects (using different indices) need to be screened. These observations indicate that the proposed widely applicable, routine, quick, easily interpreted biochemical characteristics on arrival which are currently rarely used, add significant information over standard clinical markers and may be helpful in day-to day clinical practice. When interpreting the utility of individual biochemical parameters as predictors of outcomes it should be taken into account that: (1) the presented models indicate the number of subjects with specific characteristics in which adverse outcomes may be expected, it is not an individual/personalised prediction for each patient (any person may have non-measured by the models or not known characteristics which may modify the individual risk; the preoperative signs alone may be insufficient for predicting multifactorial outcomes; the intra- and postoperative quality of care has obviously a substantial impact on the patient’s postoperative course), and (2) highly sensitive tests with low NNP are preferable to identify most cases (minimal false negatives, despite consequent risk of false positives) but tests with high specificity outweigh low NNP if the goal is to exclude most false positives (non-cases). Analysis of 2–3 tests together gives a better understanding of their prognostic gain for prediction a specific outcome; therefore, it is desirable to assess different biochemical indices as they may be complementary for prediction decisions.

In summary, the presented here simple prediction models fulfill three main criteria—acceptable discrimination power, calibration and decreased NNP. While the clinical approach to preoperative prediction may be subjective, variable among clinicians and prone to error, assessment of biochemical indices at admission provides significant objective prognostic information and improves prediction.

Furthermore, the data on prognostic value of the analysed circulating biochemical parameters all of which reflect reversible/modifiable conditions provide important knowledge regarding possible areas accessible for interventions—perioperative as well as pre-fracture. The clinical implications of considering metabolic derangements are threefold: to optimise prognostication in HF patients and avoid unnecessary perioperative assessment (i.e., cardiac), to improve clinical outcomes and to select appropriate patients for intervention prior fracture occurred.

To identify the most vulnerable HF patients at risk of an adverse outcome at admission our study focuses on two clinical characteristics (age, history of IHD) and simple biochemical indices; this approach provides useful prediction information to guide and support the shared (doctors-patient/carer) decision-making process (including selection which patient may benefit from arthroplasty surgery, internal fixation or nonoperative management) and might improve outcomes by early introducing appropriate individualised medical treatment aimed at controllable risk factors including abovementioned biochemical abnormalities. Given that developing PMI accounted for 80.3% of in-hospital mortality and was associated with elevated inflammatory response and prolonged LOS, predicting and targeting subjects with a high probability of this complication is reasonable.

Currently there is no consensus regarding perioperative management of HF patients at risk of PMI, the available data are scarce, and recommendations are controversial. Numerous studies on benefits and risks of beta-blockers in patients undergoing non-cardiac surgery, including HF, yielded controversial results [151,152,153,154,155,156,157,158,159,160,161,162]. However, reduction in cardiac complications has often been reported in patients with risk factors indicating that perioperative beta-blocker management should be individualised [163]. Similarly, perioperative use, continuation or discontinuation of antiplatelet (especially dual) therapy [164,165,166,167,168,169], antihypertensive drugs, renin-angiotensin-aldosterone system inhibitors, diuretics [170,171,172] remain debatable issues. It was reported that prophylactic nitrates [26,173], as well as alpha-2 adrenergic agonists [174] do not prevent death and cardiac complications in non-cardiac surgery. Statins are recommended preoperatively in patients with atherosclerotic CVD [12]. In patients who suffer PMI, risk-adjusted observational data suggest that aspirin and a statin can reduce the risk of 30-day mortality [175,176] and PMI, but the risk of perioperative bleeding should be fully considered before using antiplatelet and anticoagulant drugs. Obviously, the treatment decision should be discussed in a multidisciplinary way and based on patient-specific conditions (e.g., documented IHD, coronary stents, CABGs, AF, recent thromboembolic disease, inherited coagulopathy, etc.) to balance risks of possible drug-related ischemic-bleeding complications, prevent stroke, avoid intra- and postoperative hypotension, bradycardia and/or fluid overload; lower doses should be considered.

Clearly, attention to biochemical characteristics (e.g., vitamin D status, hyperparathyroidism, hypoalbuminaemia, liver function indices, etc.) should drive the much need individualised preventive management in persons at high risk of osteoporotic fracture long before the fracture occurred. Our observations on prevalence among older HF patients of alterations in serum vitamin D [51,124,126,133,177,178,179], PTH [44,45,46,47,48,49,50,51,53,133,180], albumin levels [16,51,53,54,58,80,181,182,183,184,185,186,187,188,189] and liver function [76,77,107,110,190,191,192,193,194,195,196] are consistent with reported in the literature. These data strongly suggest that these metabolic dysregulations known to be linked to osteoporosis, falls, factures and to many chronic diseases are often not remedied with standard treatment. Attempts to identify and address metabolic alterations prior to the onset of disease(s) and end-organ complications should be an important part of preventive management of osteoporotic fractures across disciplines; such approach will increase the effectiveness and quality of patient care and may reduce health care costs in the ageing population.

Our findings are particularly noteworthy in the context of current demographic and epidemiological trends. Due to increase in life expectancy (ageing population), the number of patients with osteoporosis, sarcopenia, CVD/IHD, other chronic conditions and, consequently, geriatric low energy/fragility fractures with worse prognosis after orthopaedic surgery will increase. Physicians and orthopaedic surgeons can utilise biochemical indices to predict timely adverse events. Given that metabolic alterations often present for years before becoming clinically apparent and effective interventions are usually available earlier identification of individuals at risk is particularly important.

### 4.2. Strength and Limitations

Strengths of the study include the relatively large sample of patients with any type of surgery for HF, simultaneous testing of 10 different biomarkers making direct comparisons of predictive performance of all tested indices possible, robust statistical methods used, and internal validation of the proposed models. To the best of our knowledge, this is the first study to evaluate and compare in HF patients the predictive value at admission of a wide range of commonly available different biochemical indices.

Several limitations of this study deserve comment. As it was an observational single-centre study, selective bias cannot be rule out and cause–effect relationships cannot be fully inferred from it; the data are limited to HF surgeries and may not reflect outcomes for other orthopaedic or non-cardiac surgeries. The number of deaths was relatively low. The set of biochemical indices included form only a fraction of all available metabolic variables (other characteristics may also be predictive) and no comparison with haematologic parameters of known predictive value (e.g., anaemia [197,198,199], or lymphocyte-neutrophil ratio [51,200,201], or red blood cell distribution width [202,203]) has been done. Our findings need external validation. The patients were predominantly white and of European descent, which may limit the generalisability of our results. However, the sociodemographic, clinical and laboratory characteristics, type and incidence of hospital outcomes in our cohort seems representative of the average HF population; the findings could therefore be applied at least to the whole Caucasian HF population; further studies are needed to determine whether the findings extend to other racial/ethnic groups.

In conclusion, we identified main clinical and several biochemical parameters that carry a high prognostic potential for preoperative prediction of hospital outcomes in patients with HF. The most important clinical characteristics in guiding short-term prognosis, predicting PMI and/or death include age > 80 years and history of IHD. The proposed routine easily accessible biochemical indices at admission substantially improve prognostic accuracy and prediction of outcomes and may be useful for early risk stratification and identification of the most vulnerable patients (especially among the aged >80 years with IHD) in whom appropriate treatment focused on factors allowing modification might reduce poor outcomes and improve survival.

## Figures and Tables

**Figure 1 jcm-11-06784-f001:**
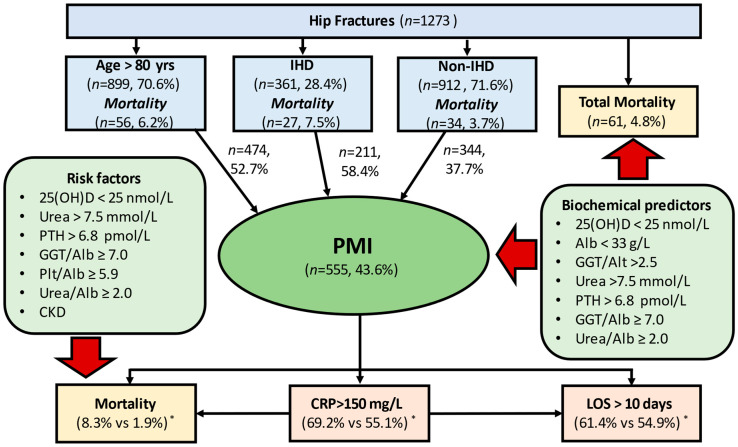
Relationships between ischaemic heart disease (IHD), postoperative myocardial injury (PMI), other adverse hospital outcomes and predictive biomarkers at admission in patients with hip fracture (HF). Among HF patients 70.6% were aged >80 years (mortality 6.2%), 28.4% have been diagnosed with IHD pre-fracture (mortality 7.5% vs. 3.7% in the non-IHD group); the total all-cause mortality 4.8%. PMI occurred in 43.6% of patients: in 58.6% among subjects with previously diagnosed IHD (11.1% had a history of AMI) and 37.7% in the non-IHD group. PMI developed in 52.7% of patients aged >80 years, in 58.4% of patients with pre-fracture known IHD, and in 37.7% of patients without IHD. PMI contributed to 80.3% (49 of 61 patients) of all deaths (mortality rate (8.8% vs. 1.9% in non-PMI) and was associated with a higher frequency of a high postoperative inflammatory response (CRP > 150 mg/L after the 3rd postoperative day: 69.2% vs. 55.1%) and prolonged hospital stay (LOS > 10 days: 61.4% vs. 54.9%). On-admission biochemical predictors of PMI and/or a fatal outcome are shown in the box on the right. Factors at admission that significantly (1.3–1.6-times) increase the risk of a lethal outcome in patients who developed PMI are listed in the left box. * Comparison with non-PMI patients.

**Table 1 jcm-11-06784-t001:** Comparison of baseline clinical characteristics and outcomes in hip fracture patients with and without ischaemic heart disease (IHD).

Variable	Total Cohort (*n* = 1273)	With IHD (*n* = 361, 28.4%)	Without IHD (*n* = 912, 71.6%)	*p* Value
Age, mean ± SD, years	82.9 ± 8.68	84.9 ± 7.18	82.2 ± 9.10	<0.001
Aged > 80 years, %	70.6	76.7	68.2	0.001
Female, %	73.5	69.5	75.0	0.028
PRCF resident, %	32.8	30.9	31.6	0.090
HF type [trochanteric], %	49.3	51.5	48.5	0.178
History of AMI, %	7.8	27.7		
Hypertension, %	55.9	66.2	51.8	<0.001
CVA, %	10.8	15.0	9.1	0.002
TIA, %	10.1	11.6	9.4	0.141
CKD, %	34.0	46.1	29.2	<0.001
COPD, %	17.2	25.2	14.0	<0.001
Anaemia, %	41.8	44.3	40.8	0.138
T2 DM, %	19.7	23.0	18.3	0.058
Dementia, %	31.6	33.0	31.0	0.273
Parkinson’s disease, %	5.0	3.3	5.6	0.058
Smoker, %	5.7	4.2	6.4	0.079
Ex-smoker, %	11.8	13.0	11.3	0.223
* Alcohol over-user, %	4.0	2.2	44.7	0.025
Walking aids user, %	37.3	41.0	35.8	0.048
In-hospital mortality, %	4.8	7.5	3.7	0.005
Myocardial injury, %	43.6	58.6	37.7	<0.001
Postoperative AMI, %	6.7	11.7	4.8	<0.001
LOS > 10 days, %	57.9	61.5	56.5	0.058
LOS > 20 days, %	22.0	25.8	20.4	0.024
CRP > 100 mg/L, %	80.9	84.2	79.7	0.037

Abbreviations: PRCF, permanent residential care facility; IHD, ischaemic heart disease; AMI, acute myocardial infarction; CKD, chronic kidney disease (estimated glomerular filtration rate < 60 mL/min/1.73 m^2^); CVA, cerebrovascular accident (stroke); TIA, transient ischaemic attack; COPD, chronic obstructive airway disease; T2DM, type 2 diabetes mellitus; LOS, length of hospital stay; CRP, C-reactive protein; * ≥3 times per week.

**Table 2 jcm-11-06784-t002:** Clinical characteristics and outcomes in hip fracture patients with and without postoperative myocardial injury (PMI).

Variable	With PMI (*n* = 555, 43.6%)	Without PMI (*n* = 718, 56.4%)	*p* Value
Age, mean ± SD, years	86.1 ± 6.82	80.8 ± 8.91	<0.001
Aged > 80 years, %	85.2	60.4	<0.001
Male, %	28.9	24.6	0.054
PRCF resident, %	38.7	28.4	<0.001
Trochanteric HF, %	48.2	49.6	0.341
History of IHD, %	37.7	20.6	<0.001
History of AMI, %	11.1	5.2	0.001
Hypertension, %	60.2	51.4	0.001
CVA, %	11.8	9.6	0.119
TIA, %	12.6	8.3	0.009
CKD, %	44.3	26.1	<0.001
COPD, %	16.5	17.5	0.347
Anaemia, %	46.0	38.4	0.005
T2DM, %	20.8	18.8	0.524
Dementia, %	38.5	26.2	<0.001
Parkinson’s disease, %	3.8	5.9	0.052
Smoker, %	4.1	6.5	0.068
Ex-smoker, %	13.0	11.7	0.286
* Alcohol over-user, %	1.9	5.4	0.001
Walking aids user, %	38.7	36.0	0.341
In-hospital mortality, %	8.8	1.9	<0.001
LOS > 10 days, %	61.4	54.9	0.013
LOS > 20 days, %	22.1	21.0	0.645
CRP > 100 mg/L, %	88.0	77.3	<0.001
CRP > 150 mg/L, %	69.2	55.1	<0.001

Abbreviations: PRCF, permanent residential care facility; IHD, ischaemic heart disease; AMI, acute myocardial infarction; CKD, chronic kidney disease (estimated glomerular filtration rate < 60 mL/min/1.73 m^2^); CVA, cerebrovascular accident (stroke); TIA, transient ischaemic attack; COPD, chronic obstructive airway disease; T2DM, type 2 diabetes mellitus; LOS, length of hospital stay; CRP, C-reactive protein; * ≥3 times per week.

**Table 3 jcm-11-06784-t003:** Prognostic value of age, presence of IHD and specific biochemical characteristics at admission for predicting in-hospital death and postoperative myocardial injury.

Variable	^1^ Total Cohort (*n* = 1273)		^2^ IHD (*n* = 361)		^3^ IHD > 80 Years of Age (*n* = 277)	
	OR (95% CI)	*p* Value	OR (95% CI)	*p* Value	OR (95% CI)	*p* Value
**In-hospital Mortality**						
Age > 80 years	4.9 (1.95–12.33)	0.001	5.0 (1.96–12.62)	0.001		
IHD	2.1 (1.24–3.51)	0.005			7.4 (2.55–21.51)	<0.001
PTH > 6.8 pmol/L	1.9 (1.06–3.25)	0.031	3.7 (1.85–7.27)	<0.001	11.8 (2.71–51.19)	0.001
25(OH)D < 25 nmol/L	2.4 (1.25–4.68)	0.009	8.0 (3.46–18.28)	<0.001	23.6 (6.73–82.54)	<0.001
25(OH)D < 50 nmol/L	1.2 (0.71–2.11)	0.473	2.7 (1.43–5.24)	0.002	8.4 (2.72–25.89)	<0.001
Albumin < 33 g/L	1.2 (0.64–2.23)	0.573	3.3 (1.46–7.64)	0.004	15.2 (3.87–59.53)	<0.001
Urea > 7.5 mmol/L	2.2 (1.34–3.95)	0.007	5.1 (2.48–1066)	<0.001	9.5 (2.8–32.47)	<0.001
GGT ≥ 26 IU	1.7 (1.01–2.94)	0.047	1.7 (0.98–2.89)	0.057	7.7 (1.72–34.33)	0.008
GGT/Albumin ratio ≥ 7	2.1 (1.21–3.69)	0.008	4.0 (1.85–8.78)	<0.001	16.5 (2.16–126.33)	0.007
GG/ALT ratio > 2.5	1.3 (0.74–2.23)	0.372	2.6 (1.129–5.44)	0.008	8.5 (2.30–31.15)	0.001
Urea/Albumin ratio ≥ 2.0	2.2 (1.22–4.07)	0.009	5.2 (2.50–11.03)	<0.001	9.2 (2.70–31.44)	<0.001
Plt/Albumin ratio ≥ 5.9	1.8 (1.05–3.16)	0.032	4.0 (1.83–8.55)	<0.001	21.8 (2.85–166.99)	<0.001
ALT ≥ 17 IU	1.1 (0.64–1.85)	0.751	1.7 (0.73–4.14)	0.216	2.42 (0.71–8.29)	0.158
**Postoperative Myocardial Injury**						
Age > 80 years	3.8 (2.83–4.99)	<0.001	3.9 (2.92–5.27)	<0.001		
IHD	2.3 (1.81–3.01)	<0.001			8.3 (5.58–12.36)	<0.001
PTH > 6.8 pmol/L	1.3 (1.02–1.68)	0.032	3.1 (2.20–4.35)	<0.001	8.3 (5.01–13.73)	<0.001
25(OH)D < 25 nmol/L	0.97 (0.65–1.44)	0.869	3.9 (1.92–8.03)	<0.001	18.0 (7.00–46.26)	<0.001
25(OH)D < 50 nmol/L	1.04 (0.80–1.35)	0.768	2.3 (1.55–3.33)	<0.001	10.4 (5.78–18.83)	<0.001
Albumin < 33 g/L	0.8 (0.59–1.09)	0.162	1.8 (1.07–3.02)	0.026	6.6 (3.37–13.08)	<0.001
Urea ≥ 7.5 mmol/L	1.5 (1.21–1.99)	0.001	4.4 (3.12–6.36)	<0.001	12.8 (7.71–21.31)	<0.001
GGT ≥ 26 IU	0.98 (0.76–1.25)	0.861	2.5 (1.74–3.64)	<0.001	9.1 (5.07–16.16)	<0.001
GGT/Albumin ratio ≥ 7.0	1.04 (0.81–1.33)	0.753	2.4 (1.66–3.33)	<0.001	9.9 (5.51–17.62)	<0.001
GG/ALT ratio ≥ 2.5	0.99 (0.761.30)	0.965	2.1 (1.44–3.14)	<0.001	11.4 (6.25–20.71)	<0.001
Urea/Albumin ratio ≥ 2.0	1.50 (1.19–1.96)	0.001	4.5 (3.18–6.39)	<0.001	12.6 (7.62–20.89)	<0.001
Plt/Albumin ratio ≥ 5.9	0.89 (0.70–1.14)	0.351	2.1 (1.45–3.10)	<0.001	6.1 (3.50–10.53)	<0.001
ALT ≥ 17 IU	0.95 (0.74–1.21)	0.661	2.8 (1.96–3.88)	<0.001	7.5 (4.05–13.77)	<0.001

Abbreviations: OR, odds ratio; CI, confidence interval; IHD, ischaemic heart disease; PTH, parathyroid hormone; 25(OH)D, 25 hydroxy vitamin D; GGT, gamma-glutamyl transferase; ALT, alanine aminotransferase; Plt, platelets. ^1^ adjusted for age, gender and all clinical variables which were significantly associated with hospital mortality on univariate analyses; ^2^ comparison IHD patients with the rest of the cohort, adjusted for age (as a continues variable), gender and clinical variables significantly associated with hospital mortality or postoperative myocardial injury on univariate analyses (Table 1 and Table 2); ^3^ comparison of IHD patients aged >80 years and younger than 80 years with and without the analysed characteristic.

**Table 4 jcm-11-06784-t004:** Summary of performance parameters of biomarkers at admission to predict in-hospital mortality and/or postoperative myocardial injury in aged (>80 years) hip fracture patients with IHD.

Biomarker	AUC (95% CI)	Sensitivity (%)	Specificity (%)	Accuracy (%)	PPV (%)	NPV (%)	LR+	LR−	NNP	* Calibration, chi^2^ (*p* Value)
**In-Hospital mortality**										
25(OH)D < 25 nmol/L	0.803 (0.671–0.934)	69.2	91.3	90.3	28.1	98.4	7.946	0.337	3.77	9.11 (0.6113)
Albumin < 33 g/L	0.789 (0.649–0.929)	72.7	85.1	84.6	16	98.8	4.866	0.321	6.76	6.97 (0.8012)
GGT/ALT > 2.5	0.742 (0.627–0857)	78.6	69.8	70.2	22.4	98.4	2.6	0.307	4.81	7.44 (0.7524)
25(OH)D < 50 nmol/L	0.739 (0.644–0.830)	80	67.7	68.6	16.3	97.7	2.478	0.295	7.14	20.39 (0.0401)
Urea > 7.5 mmol/L	0.729 (0.653–0.803)	87	58.7	60.4	12.1	98.5	2.105	0.222	9.52	4.89 (0.9363)
PTH > 6.8 pmol/L	0.725 (0.658–0.792)	90.9	54.1	56.4	12	98.9	1.979	0.168	9.17	8.20 (0.6955)
GGT/Albumin ≥ 7.0	0.725 (0.659–0.790)	94.1	50.8	53.5	11.3	99.2	1.912	0.116	9.52	4.38 (0.9574)
Urea/Albumin ≥ 2.0	0.7216 (0.649–0.794)	87.5	56.8	58.7	11.9	98.6	2.027	0.22	9.52	4.74 (0.9433)
Plt/Albumin ≥ 5.9		94.1	57.7	59.9	19.8	99.3	2.225	0.102	5.24	8.43 (0.6747)
IHD + Age > 80 years	0.700 (0.634–0.765)	86.7	53.3	55	9.4	98.6	1.854	0.25	12.5	1.26 (0.8681)
GGT ≥ 26 IU	0.694 (0.609–0.780)	88.2	50.6	53.1	11	98.4	1.787	0.232	10.64	2.71 (0.9941)
ALT ≥ 17 IU	0.607 (0.464–0.751)	66.7	54.8	55.4	7.5	96.8	1.475	0.608	23.26	14.82 (0.1908)
Age > 80 years	0.637 (0.584–0.691)	91.8	30.4	33.4	6.2	98.7	1.32	0.269	20.41	0.29 (0.5876)
IHD	0.591 (0.519–0.663)	44.3	72.4	71.1	7.5	96.3	1.606	0.769	26.3	0.17 (0.6766)
**Postoperative myocardial injury**										
Urea > 7.5 mmol/L	0.782 (0.738–0.825)	87.7	78.8	82.2	71.7	91.3	4.131	0.156	1.59	5.47 (0.9063)
Urea/Albumin ≥ 2.0	0.780 (0.737–0.823)	79.2	76.8	77.8	69.8	84.5	3.416	0.271	1.84	6.66 (0.8261)
GGT/ALT > 2.5	0.760 (0.705–0.816)	67.4	84.6	79	68.2	84.2	4.382	0.385	1.91	13.44 (0.2654)
25(OH)D < 50 nmol/L	0.757 (0.702–0.812)	68.8	82.6	77.7	68.1	83	3.946	0.378	1.96	6.46 (0.841)
GGT/Albumin ≥ 7.0	0.755 (0.703–0.806)	80.7	70.2	74.7	67.2	82.8	3.708	0.275	2	10.14 (0.5176)
PTH > 6.8 pmol/L	0.742 (0.694–0.789)	76.6	71.6	73.7	65.6	81.3	2.703	0.326	2.13	12.30 (0.3417)
IHD + Age > 80 years	0.741 (0.703–0.778)	77.7	70.4	73.4	64.3	82.2	2.626	0.316	2.15	1.88 (0.758)
Plt/Albumin ≥ 5.9	0.711 (0.654–0.768)	71	71.3	71.2	60.7	79.7	2.47	0.407	2.27	9.28 (0.5960)
25(OH)D < 25 nmol/L	0.665 (0.608–0.723)	36.1	97	80.7	81.3	80.6	11.856	0.659	1.62	12.94 (0.2970)
Albumin < 33 g/L	0.656 (0.593–0.720)	53.1	90.7	83.7	56.5	89.4	5.678	0.518	2.18	12.92 (0.2986)
ALT ≥ 17 IU	0.732 (0.672–0.792)	74.7	71.6	72.9	63.1	81.4	2.635	0.353	2.25	4.71 (0.9446)
GGT ≥ 26 IU	0.748 (0.695–0.802)	85.7	70.7	77.1	68.7	86.8	2.927	0.202	1.8	13.48 (0.2629)
Age > 80 years (total cohort)	0.624 (0.600–0.647)	85.2	39.6	59.4	52.1	77.6	1.409	0.375	3.37	0.00 (0.9465)
IHD (total cohort)	0.593 (0.593–0.623)	37.7	79.4	61.2	58.6	62.3	1.832	0.784	4.78	0.68 (0.4088)

Abbreviations: OR, odds ratio; CI, confidence interval; AUC, area under the curve (receiver operating characteristic); PPV, positive predictive value; NPV, negative predictive value; LP+, positive likelihood ratio; LP− negative likelihood ratio; NNP, number of patients needed to be examined for correct prediction; Plt, platelets; none of CAD survivors aged > 80 years had Plt/Albumin ratio ≥ 5.9 (therefore AUC could not be calculated). * Calibration, Hosmer-Lemeshow goodness-of-fit test (for full models).

## Data Availability

All data analysed as part of the study are included.

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
