# Peer review of "Comparison of Prognostic Value of 10 Biochemical Indices at Admission for Prediction Postoperative Myocardial Injury and Hospital Mortality in Patients with Osteoporotic Hip Fracture"

_jcm, 2022, doi:10.3390/jcm11226784_

Round 1

Reviewer 1 Report

This article took the common osteoporotic hip fracture as the research object. The researchers explored the relationship between 10 biochemical indicators selected by themselves in predicting postoperative myocardial injury and in-hospital mortality in patients with this disease.They believed that the biochemical indicators easily obtained at admission among HF patients, especially among patients with IHD over 80 years old, significantly improved the prediction of hospital outcomes, and that these indicators could be used for prediction.The design idea was relatively new, the scheme was logical, the research sample size was large, the results were true and reliable, and the conclusions were basically correct. However, this article needs to be properly modified or explained.

(1)In the introduction, only the references and 10 indicators selected by the researchers were given, but the researchers did not explain why only these 10 indicators were selected, please give reasons;

(2)In the methodology, specific inclusion criteria and exclusion criteria should be given;

(3)In the methodology, please give the ethical approval number;

(4)In the discussion, the researchers only discussed the several indicators which associated with their conclusion, but they did not explain how and why did the indicators work, please give explaination in this part.

(5)There were too many references, so it is recommended to delete them appropriately.

Author Response

Responses to the Reviewers

To Reviewer 1

We are very grateful for your time reviewing our manuscript and the helpful comments. We would like to present the changes which have been made and clarify some issues.

(1)In the introduction, only the references and 10 indicators selected by the researchers were given, but the researchers did not explain why only these 10 indicators were selected, please give reasons.

In this study we focussed on predictive value of 10 routine biochemical indices in poor outcomes in patients with hip fracture. We acknowledge that the set of biochemical indices included form only a fraction of all available metabolic, haematologic, immunologic, etc. variables and addressed this issue in the ”Limitations” (p.21, lines 854-859). In the introduction (p.2, lines 88-90) we have mentioned that the “chosen biomarkers are easily obtained parameters known to be associated with  chronic systemic diseases, in particular CVDs/IHD and osteoporosis, and being factors allowing modification”. Moreover,” these biomarkers have not been systematically investigated to predict in  HF patients at admission the risk of development postoperative myocardial  injury (PMI) or death.” p.2, lines 96-98).

(2)In the methodology, specific inclusion criteria and exclusion criteria should be given.

Thank you for this advice. The paragraph (p.3, lines108-114) has been rewritten and the following statement added:

“Inclusion criteria were as follows: (1) a definite diagnosis of hip fracture (intracapsular [cervical] or trochanteric) by imaging, (2) surgical HF repair, (3) age ≥60 years, (4) complete clinical and laboratory data. Exclusion criteria were: (1) subtrochanteric fracture, (2) medium- or high- energy trauma fracture (fall from height, car accident, etc.); (3) multiple fractures or polytrauma, (4) pathological fracture (malignant tumour)”

(3)In the methodology, please give the ethical approval number

Requested information has been added (p.4, line 170):    

“The study was approved  by the Australian Capital Territory Research Ethics Committee (ETHLR.18.085; REGIS Reference 2020/ETH02069)” .

(4)In the discussion, the researchers only discussed the several indicators which associated with their conclusion, but they did not explain how and why did the indicators work, please give explanation in this part.

Because the complexity of pathophysiologic causes and effects of different biochemical parameters and their interactions have been discussed (at least partially) in our previous publications (references 17, 47, 51, 121, 122, 200) and addressed in numerous papers cited in this article we limited the discussion with a short integrative description/explanation of underlying biological mechanisms (p. 18, lines 664-683):

“Metabolic indicators integrate various genetic, lifestyle and environmental effects, reflect ongoing physiological processes in multiple organs, and, therefore, provide an insight into disease pathophysiology acting as powerful characteristics of an individual’s health status. Metabolic dysregulations are implicated in the pathogenesis and severity  of numerous human pathologies, including osteoporosis, atherosclerosis, IHD (reflecting systemic connectivities), and, consequently, underly the adverse  outcomes in HF patients. In HF patients, alterations in biochemical indices, IHD and advanced age interact to increase risks of developing PMI, new AMI, high inflammatory response, prolonged LOS and mortality above the effect of  each individual condition, as demonstrated in this and our previous studies  [17, 45, 51, 110, 120-122]. Our findings that on-arrival insufficient 25(OH)D levels and/or elevated PTH predict PMI and/or death are in line with long-standing evidence that  the physiological effects of vitamin D and PTH extend beyond calcium homeostasis and bone mineralisation and consistent with many previous reports  [17, 123-126]. Hypovitaminosis D is also known to correlate with higher inflammatory response [124, 127-132], which together with increased PTH levels (a factor associated with multiple postoperative complications in HF patients [45, 133] and/or PMI significantly increases the risk of poor outcomes”.

(5)There were too many references, so it is recommended to delete them appropriately.

We appreciate your opinion. We have also had some concerns about the length of the paper and the number of references. However, because the published literature (often in journals related to different disciplines) on each of the topics is often controversial, we think that it is important to present to the interested reader a comprehensive spectrum of current data and views, which, obviously, requires extended references.

Reviewer 2 Report

Thank you for the opportunity to review this very interesting and potentially very practical manuscript. The study presents the assessment of the prognostic values of selected biochemical parameters in terms of the prognosis of in-hospital mortality and/or postoperative myocardial injury. The study is prospective, with a large population of hip fractures patients, and evaluates a relatively large number of biochemical parameters. Some of the biochemical parameters may be modifiable, which provides the basis for further research or modifying these variables will improve the prognosis in patients with hip fracture.

1.   I propose to round the AUC values to 3 decimal places.

2.       Page 3 line 101 - close the parenthesis.

3.       Page 3 line 115 - please describe better the validation cohort from which group of patients they were extracted. Please describe why you used validation cohort in this study.

4.       Page 4 methodology - what if the patient had cTnI> 20 ng/L before surgery? In the case of a constant elevated concentration of cTnI before and after the procedure, was PMI also diagnosed (e.g. before and after the procedure, 30 ng/L)?

5.       Page 5 line 205 - t2DM and Parkinson's disease did not reach statistical significance.  

6.       Page 5 line 210 - please specify p for previous AMI (contrast + 8.1%).

7.       Page 5 line 223 – please specify OR, 95% CI, and p for postoperative AMI.

8.       Page 6 line 233 – sex was not statistically significant.

9.       Page6 line 238 – please specify p in alcohol over-users.

10.   Page 6 line 238 – “current smokers” was not statistically significant.

11.   Page 6 line 239 – Parkinson's disease was statistically insignificant.

12.   Page 10 line 324-356 – it is not clear whether additional variables are added to the model or whether they are models with one additional variable. Please consider presenting the models in the tables, it would be easier to read. This part of the article is, in my opinion, one of the most interesting pieces of information.

13.   Page 11 line 358-370 – two multivariate logistic regression models were presented, but it is clear what these models assessed. The first, described as "mortality at admission", indicates that someone died on admission to the hospital, this seems to be an unfortunate sentence; My guess is that these are parameters determined on admission to the hospital. I am asking for a clearer description of the model. Second model for "hospital death" ... I don't understand why there is a new model similar to the first "mortality at admission" or is it a model that presupposes the occurrence of PMI? If so, it should be described in a more obvious way. Presenting and describing the models in tables would be clearer.

14.   Table 4 – the description of the table shows that the AUC values of individual parameters for "In-Hospital mortality" and "PMI" for the group of patients> 80 years old with IHD are presented in it. Why then do you present AUC for age> 80 years, presence of IHD and> 80 years + IHD? The description of the table shows that the whole group has variables> 80 years and IHD. Doing AUC for these variables (> 80 years and IHD) would make sense for the entire study group with and without these variables. From the text in line 423 it appears that this is an extracted group of patients with> 80 years and IHD.

15.   Page 12 line 390-393 – is the 25(OH)D model composed of> 80years + IHD + 25(OH)D or is it AUC for 25(OH)D only for the entire study population? This question applies to all models in this line.

16.   Page 13 line 401 – you write "predictive specificity", please make it clear what parameter you are describing, because then you give the specifitity value and then accuracy. Please do not confuse these parameters.

17.   Page 13 line 447 - the sentence "56 vs. 61 "is inadequate to the data. More adequate would be “56 vs. 5” but I would suggest deleting this sentence.

18.   Please consider deleting repetitions in the text that are already contained in the tables. There are many of them. Without these repetitions, the text would be clearer.

19.   Page 14 line 492-502 - this is a very important part of this article. Please consider reporting the AUC for death in the group of patients with PMI for the parameters shown.

20.   Page 16 line 602 – lack of „with” after “associated”.

21.   Page 17 line 628 – “4.6-fild”? or fold?

Author Response

Responses to the Reviewers

To Reviewer 2

We are most grateful for your hard work and time reviewing our manuscript and for your important comments. We would like to present the changes we made following your recommendations and clarify some issues.

  1. I propose to round the AUC values to 3 decimal places.

Changes done in both the text and Table 4.

  1. Page 3 line 101 - close the parenthesis.

Thank you, done.

  1. Page 3 line 115 - please describe better the validation cohort from which group of patients they were extracted. Please describe why you used validation cohort in this study.

The validation cohort was a separate group of patients with hip fracture admitted after those in the derivation group; this information is added.

Rigorous validation is needed to reliably assess the prognostic and predictive values of the proposed models. We have been able to perform only an internal validation. Our findings are based on a single-centre study and require external validation to confirm their clinical value.  

  1. Page 4 methodology - what if the patient had cTnI> 20 ng/L before surgery? In the case of a constant elevated concentration of cTnI before and after the procedure, was PMI also diagnosed (e.g. before and after the procedure, 30 ng/L)?

In this study PMI was diagnosed only in patients who developed post- (not pre-) surgical cTnI elevation.

  1. Page 5 line 205 - t2DM and Parkinson's disease did not reach statistical significance.

You are absolutely right, the p values for these two diseases are of borderline significance  and, therefore, shown in brackets.

  1. Page 5 line 210 - please specify p for previous AMI (contrast + 8.1%).

Thank you, data added: p<0.001.

  1. Page 5 line 223 – please specify OR, 95% CI, and p for postoperative AMI

Thank you, missed data inserted: (OR 2.4, 95%CI 1.98-4.02, p<0.001).

  1. Page 6 line 233 – sex was not statistically significant.

Indeed, the p value was 0.054 indicating only borderline (“approaching”) significance; in other words, it  was close enough to 0.05 to be worth commenting on. Of note, the popular statistical significance using a p of 0.05 is just an arbitrary cut-off .

  1. Page 6 line 238 – please specify p in alcohol over-users.

Thank you, missed data inserted: p=0.025.

  1. Page 6 line 238 – “current smokers” was not statistically significant.

The p value  shown clearly suggests that the statistical significance is only borderline.

  1. Page 6 line 239 – Parkinson's disease was statistically insignificant.

The p value is given in brackets (p=0.052), and the reader can come to his own decision regarding the importance of the borderline significance (when the association is not that strong enough to reach statistical significance)

  1. Page 10 line 324-356 – it is not clear whether additional variables are added to the model or whether they are models with one additional variable. Please consider presenting the models in the tables, it would be easier to read. This part of the article is, in my opinion, one of the most interesting pieces of information.

Many thanks for your comment and valuable advice. As shown in Table3 all these models   include only one additional variable, text corrected (p.10, line 325 of 1st PDF draft).

  1. Page 11 line 358-370 – two multivariate logistic regression models were presented, but it is clear what these models assessed. The first, described as "mortality at admission", indicates that someone died on admission to the hospital, this seems to be an unfortunate sentence; My guess is that these are parameters determined on admission to the hospital. I am asking for a clearer description of the model. Second model for "hospital death" ... I don't understand why there is a new model similar to the first "mortality at admission" or is it a model that presupposes the occurrence of PMI? If so, it should be described in a more obvious way. Presenting and describing the models in tables would be clearer.

Thank you, to avoid confusion the text was modified:

In the total HF population, multivariate logistic regression model (included all variables associated with in-hospital death on univariate analysis) revealed as significant independent predictors of mortality at admission the following: age >80 years, PTH >6.8 pmol/L, urea >7.5 mmol/L and GGT/Albumin≥7.0; this model yielded AUC of 0.725 (95%CI 0.663 - 0.788). The multivariate logistic regression for hospital death based on the same approach and development of PMI (postoperative cTnI rise) demonstrated in addition to abovementioned characteristics as independent predictors also vitamin D <25mmol/L and PMI; this model improved the prediction a fatal outcome and yielded AUC of 0.767 (95%CI 0.710– 0.824). The independent preoperative predictors of PMI (in the total HF cohort) were history of IHD, age >80 years, PTH >6.8 pmol/L, urea >7.5 mmol/L and male sex, AUC 0.700 (95%CI 0.671 – 0.730).  

  1. Table 4 – the description of the table shows that the AUC values of individual parameters for "In-Hospital mortality" and "PMI" for the group of patients> 80 years old with IHD are presented in it. Why then do you present AUC for age> 80 years, presence of IHD and> 80 years + IHD? The description of the table shows that the whole group has variables> 80 years and IHD. Doing AUC for these variables (> 80 years and IHD) would make sense for the entire study group with and without these variables. From the text in line 423 it appears that this is an extracted group of patients with> 80 years and IHD.

We greatly appreciate your opinion. The reason to present in this Table separate data on patients aged>80 years, individuals with IHD and the combination of these two variables was to show the usefulness of predictive value of the proposed individualised approach (based on three characteristics - two clinical and one biochemical) in comparison with the commonly used  analysis of a total cohort.

  1. Page 12 line 390-393 – is the 25(OH)D model composed of> 80years + IHD + 25(OH)D or is it AUC for 25(OH)D only for the entire study population? This question applies to all models in this line.

Thank you, the issue addressed, text was clarified.

  1. Page 13 line 401 – you write "predictive specificity", please make it clear what parameter you are describing, because then you give the specificity value and then accuracy. Please do not confuse these parameters.

Thank you, text corrected, the word “predictive” deleted.

  1. Page 13 line 447 - the sentence "56 vs. 61 "is inadequate to the data. More adequate would be “56 vs. 5” but I would suggest deleting this sentence.

Following your advice data deleted.

  1. Please consider deleting repetitions in the text that are already contained in the tables. There are many of them. Without these repetitions, the text would be clearer.

The text has been modified partially. We take into account that for a busy reader (e.g., orthopaedic surgeon) it will be important to have the evidence straight away in the text before he/she is exploring the content of the tables.

  1. 19. Page 14 line 492-502 - this is a very important part of this article. Please consider reporting the AUC for death in the group of patients with PMI for the parameters shown.

 All AUCs for the reported models are presented in Table 4.

  1. Page 16 line 602 – lack of „with” after “associated”.

Thank you, text was corrected.

Page 16 Lines 595-608 should be placed under Figure 1 (as shown). “Among HF patients 70.6% were aged >80 years (mortality 6.2%), 28.4% have been diagnosed with IHD pre-fracture (mortality 7.5% vs. 3.7% in the non-IHD group); the total all-cause mortality 4.8%. PMI occurred in 43.6% of patients: in 58.6% among subjects with previously diagnosed IHD (11.1% had a history of AMI) and 37.7% in the non-IHD group. PMI developed in 52.7% of patients aged>80 years, in 58.4% of patients with pre-fracture known IHD, and in 37.7% of patients without IHD. PMI contributed to 80.3% [49/61] of all deaths (mortality rate (8.8% vs.1.9% in non-PMI) and was associated with a higher frequency of a high postoperative inflammatory response (CRP>150mg/L after the 3rd postoperative day: 69.2% vs. 55.1%) and prolonged hospital stay (LOS>10 days: 61.4% vs. 54.9%). On-admission biochemical predictors of PMI and/or a fatal outcome are shown in the box on the right. Factors at admission that significantly (1.3-1.6 -times) increase the risk of a lethal outcome in patients who developed PMI are listed in the left box.”

  1. Page 17 line 628 – “4.6-fild”? or fold

Thanks, corrected: 4.6-fold.

Many thanks for your helpful comments, advice and indicating the typos and errors, all of which have been corrected.